# Usability and optimization of online apps in user's context

M. Waseem Iqbal[1,2], Khlood Shinan[3], Shahid Rafique Shahid Rafique[4], Abdullah Alourani[5], M. Usman Ashraf[6] and Nor Zairah Ab Rahim[1]

[1] Faculty of Artificial Intelligence (FAI), Universiti Teknologi Malaysia, Jalan Sultan Yahya Petra, Malaysia
[2] Department of Software Engineering, Superior College, Lahore, Lahore, Punjab, Pakistan
[3] Department of Computers, College of Engineering and Computers in Al-Lith, Umm Al-Qura University, Makkah, Saudi Arabia
[4] Department of Computer Science, Superior College, Lahore, Lahore, Punjab, Pakistan
[5] Department of Management Information Systems, College of Business and Economics, Qassim University, Buraydah, Saudi Arabia
[6] Department of Computer Science, GC Women University Sialkot, Sialkot, Pakistan

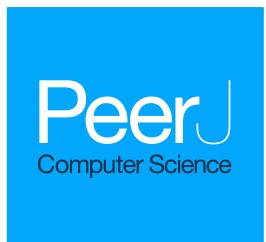

Corresponding authors
Abdullah Alourani,
ab.alourani@qu.edu.sa
M. Usman Ashraf,
usman.ashraf@gcwus.edu.pk

## ABSTRACT

The OptiFlow framework introduces a novel approach for enhancing usability evaluation and optimization known as OptiFlow. This framework combines heuristic evaluation with a web-based platform to provide a comprehensive method for assessing and optimizing user experiences in online applications. The architecture of OptiFlow incorporates key components, including the user, website, web service, and library, enabling seamless interaction and data exchange. A set of 240 usability guidelines, derived from a multidisciplinary expert collaboration, are systematically categorized into 15 usability categories, aligned with established design principles. Guidelines within OptiFlow are assigned implementation levels: "Green" for easily implementable guidelines, "Amber" for moderately complex ones, and "Red" for highly complex guidelines. These levels prioritize tasks based on complexity and feasibility. The framework's integration of guidelines into a structured SQL database simplifies implementation challenges, and the "execute" function systematically assesses website adherence to guidelines, resulting in True, False, or Null outcomes. Usability assessment outcomes are presented through categorized and prioritized data views for each implementation level, allowing stakeholders to address high-priority concerns efficiently. The OptiFlow framework represents an innovative approach to usability evaluation, fostering enriched user experiences and finely tuned digital interfaces. Future advancements may include additional rule types and the integration of advanced technologies for tackling intricate usability challenges. Ultimately, OptiFlow paves the way for proactive user experience enhancement and digital interface optimization in an ever-evolving digital landscape.

## INTRODUCTION

The advent of the internet has revolutionized various aspects of human life, including communication, commerce, and entertainment. One of the most significant outcomes of this digital era is the proliferation of web-based applications, which have become an integral part of our daily lives (*Cen et al., 2023*). These applications, ranging from

e-commerce platforms and social media networks to productivity tools and content streaming services, have transformed the way the studyinteract with information and engage with online services (*Carrera-Rivera, Larrinaga & Lasa, 2022*). Internet connectivity plays a vital role in the usability and optimization of web-based applications. The evolution of internet technologies has led to faster and more reliable connections, enabling users to access and interact with online applications seamlessly (*Vlachogianni & Tselios, 2022*). However, with the introduction of broadband connections, users gained access to high-speed internet, enhancing their online experiences. The transition from 2G to 3G and, subsequently, 4G and 5G mobile networks has further revolutionized internet connectivity (*Macakoğlu, Peker & Medeni, 2022*). These advancements have not only improved download and upload speeds but also reduced latency, making real-time interactions with web-based applications smoother and more responsive. With faster and more stable connections, users can engage with online applications more effectively, leading to enhanced usability and satisfaction (*Monzón, Angeleri & Dávila, 2021*). Major players in the web browser market, such as Google Chrome, Mozilla Firefox, and Microsoft Edge, continuously release updates to improve their speed, stability, and user interface, providing a smoother browsing experience. Modern web browsers incorporate features that enhance the usability of web-based applications. (*Xiong, Huihui & Yu, 2021*). They implement security protocols such as HTTPS, which ensures secure communication between the user's device and the web application's server. This helps in building trust and confidence among users, ultimately contributing to the overall usability and optimization of web-based applications (*Namoun, Alrehaili & Tufail, 2021*). The continuous evolution of programming languages and web development frameworks has facilitated the creation of more sophisticated and optimized web applications. Frameworks such as React, Angular, and Vue.js provide developers with a set of tools, libraries, and best practices that streamline development, improve code reusability, and optimize performance (*Perlman, 2021*). These frameworks enable the creation of responsive, interactive, and scalable web applications that cater to the diverse needs of users. Additionally, advancements in JavaScript frameworks and libraries have led to the development of Single-Page Applications (SPAs) (*Macakoğlu & Peker, 2022*). Mobile optimization is crucial for the usability and optimization of web-based applications as more users access the internet and interact with applications through their smartphones and tablets (*Akgül, 2021*). By providing a seamless and optimized experience across different devices, web applications can attract and retain mobile users while ensuring that they can accomplish tasks efficiently. By analyzing user behavior, preferences, and patterns, applications can personalize content, recommendations, and interactions. Personalization not only enhances usability but also increases user engagement and satisfaction (*Xinghai, 2023*). Data-driven insights enable web applications to understand user preferences and tailor the experience accordingly. By tracking user interactions, applications can identify patterns and trends, providing valuable insights into user behavior (*Kuparinen, 2023*). Personalization goes beyond basic customization by adapting the application experience to individual users. This level of personalization enhances the usability of web applications by reducing information overload and providing users with relevant and valuable content.

Machine learning algorithms play a significant role in data-driven insights and personalization (*Novák et al., 2023*). These machine learning algorithms analyze large datasets to identify patterns, to make predictions, and to generate personalized recommendations. Continuous user interactions, machine learning models can adapt and improve over time, providing increasingly accurate and valuable insights for optimizing web-based applications. User-centered design (UCD) is an essential approach in the development of web-based applications to ensure usability and optimization. The UCD process typically includes activities such as user research, persona development, user journey mapping, and usability testing (*Franjić, Grbac & Brumen, 2023*).

User research involves gathering insights into users' behaviors, motivations, and pain points through techniques like surveys, interviews, and observations. Usability testing is a crucial aspect of UCD, where representative users perform tasks on the application while researchers observe and gather feedback. This helps in identifying usability issues, understanding user satisfaction, and making informed design decisions (*Song et al., 2023*). By incorporating UCD principles throughout the development process, web-based applications can be optimized to meet user needs, resulting in improved usability and user satisfaction. Performance optimization is crucial for web-based applications to deliver fast user experiences. Slow page load times and delays in response can significantly impact user satisfaction and engagement. Therefore, developers employ various techniques to optimize the performance of web applications (*Germann et al., 2022*).

Some performance optimization techniques are:

- Minifying and compressing JavaScript, CSS, and HTML files to reduce file sizes and improve loading times.
- Caching static resources, such as images and CSS files, to reduce server requests and improve page load speed.
- Implementing lazy loading, which loads only the necessary content or images as the user scrolls, reducing the initial load time.
- Optimizing database queries and server-side code to improve overall application performance.
- Utilizing content delivery networks (CDNs) to distribute resources across multiple servers geographically, reducing latency and improving response times.
- By optimizing the performance of web-based applications, developers can provide users with a fast and seamless experience, leading to increased user satisfaction and engagement (*Riasat et al., 2023*).

In this ever-evolving digital landscape, the success of online applications largely hinges on their usability and optimization. Usability refers to the ease where users can interact with a system or application. While optimization involves to enhance the performance and efficiency of application to meet user needs. The usability and optimization of web-based applications play a critical role in attracting and retaining the users, enhancing their satisfaction, and ultimately achieving the goals of the application's developers.

- Investigate the usability principles and best practices in web-based applications.
- Explore optimization strategies to enhance the performance and efficiency of web-based applications.
- Examine the importance of accessibility and propose methods to ensure proper guidelines.
- Provide practical recommendations and strategies for improving the usability and optimization of web-based applications.
- Contribute to the existing body of knowledge in the field of user experience in web-based applications (*Xiong, Huihui & Yu, 2021*; *Franjić, Grbac & Brumen, 2023*; *Riasat et al., 2023*).

Below are some research questions that will guide our research in the context of web app usability and optimizations. To address the pressing issues that afflict the world of digital user experiences, receive these questions must receive prompt attention.

Q1: What does it mean to standardize the usability evaluation of web sites across different applications?

Q2: Finally, the research question of this article is: "What are the usability evaluation framework essentials that could be used globally?"

Q3: "What is actually possible to achieve when using heuristic evaluation and how can the results be further used to assess usability in different contexts?"

Q4: "Where particularly does Optiflow provide benefits over other frameworks in enhancing the user experience?" In this ever-evolving digital landscape, the success of online applications largely hinges on their usability and optimization.

# LITERATURE REVIEW

Designing products with the end user in mind offers numerous advantages and enhances perceived usability. However, the current state of product creation poses challenges, is costly, and lacks efficiency due to the requirement of comprehensively understanding the human-machine system. Consequently, the importance of prioritizing the user remains undervalued. Customers highly appreciate products that are tailored specifically to their needs. The key element in this process is the designer's conceptual model and its realization (*Akgül, 2022*). The Usability Study Evaluation Model (USE-Model) aims to achieve a better understanding of the product's capabilities. Thus, it is crucial to comprehend how users interact with the technical system and their surrounding environment. The process involves incorporating the model into the development process, utilizing insights from usability studies. This process comprises three phases: detection, analysis, and assessment. These methods provide a logical progression for designers, starting with the analysis of field data and progressing to laboratory testing and evaluation of promising features. Ultimately, the USE Model and USE Process serve as facilitators in developing customer-centric products (*Abdulrahman, Rawf & Gahfoor, 2022*).

The usability of a software product in adapting to user needs and preferences across different settings is a vital indicator of its effectiveness. Human computer interaction (HCI) is a significant research area which focuses on the user's context to access effectiveness, efficiency and satisfaction of users. E-commerce discussions have become commonplace, referring to conducting business online. Customer retention is a primary focus for most internet businesses, and usability plays a crucial role in the success of e-commerce websites. Previously, extensive efforts have focused on identifying the characteristics and features that contribute to efficient e-commerce websites. This study aims to achieve three objectives: evaluate the usability of Flipkart, Amazon, Daraz, Alibaba, and Walmart's e-commerce websites through a survey, compare their usability using Nielsen's usability heuristics, and identify the most efficient website for specific goals (Shah, 2022). The study focuses on designing, developing, and evaluating an e-commerce web application, addressing critical usability issues identified by HCI design patterns. Effectiveness testing measures user satisfaction and interface effectiveness. Questionnaires are used to assess the usability and functionality of the websites. Consumers can choose to create an account at one of the five online stores and begin shopping (Maidom et al., 2022).

The research team behind this project aims to gain a deeper understanding of the effectiveness and importance of Turkey's central government web services. Throughout the research, the authors evaluated 112 E-Government Gateway websites. Several criteria were employed to assess these websites, including user-friendliness, public engagement, trustworthiness, transparency, accountability, conversation, service quality, and integrity. Additionally, they examined the efficiency of official Turkish websites to ascertain their adherence to international standards (Duran et al., 2023). Usability was measured using various metrics, some of which you may already be familiar with, such as bounce rate, design optimization, Google page rank, download time, mobile compatibility, markup validation, response time, page size, traffic rank (Turkey), and traffic rank (Global). Furthermore, they conducted a readability analysis of all Turkish government websites at a national level, utilizing the Gunning Fog Index, Flesch-Kincaid Grade Level, and Flesch-Kincaid Reading Ease. The findings indicate that Turkey's online government services are deficient in quality. Based on the study's results, it is evident that these websites exhibit subpar functionality and speed. The analysis also uncovered concerns related to usability and security. The implications of these findings for theory, practice, and policy are significant (Gruzdo et al., 2023).

Hence, the availability of internet access is crucial for schools to operate effectively. Students heavily rely on university websites as an essential tool for their academic tasks. However, these websites often encounter multiple usability issues that can hinder visitors' experience. To address this matter, the author conducted a study using user-based evaluation and questionnaire methods, focusing on the websites of three Kurdistan Regional Government (KRG) organizations in Iraq that obtained the lowest rankings on the Ranking Web of Organizations (Webometrics) (Alghamdi et al., 2022). Thirty employees participated in answering ten survey questions and engaging in six user-based methodological exercises. The research findings reveal that the websites of Raparin

University, Halabja University, and Garmian University achieved rankings of 86.7%, 79.5%, and 61.1%, respectively. Furthermore, the customer satisfaction rate for Raparin University is 3.59, while Halabja University and Garmian University received ratings of 3.24 and 3.01, respectively (*Sundqvist & Zurawska, 2023*).

The objective of this research is to identify the key factors that can significantly enhance the user experience (UX) of a redesigned website. To gain a deeper understanding of the UX, they examined ten popular websites. Additionally, a small-scale survey was conducted to gather insights on user satisfaction levels, overall perceptions, and the UX of the website. The authors compared this information with data from three of the most widely used websites to identify any notable shifts in UX (*Tingmin & Zihe, 2023*). While they were able to categorize the features that contribute to an improved UX, they also found that these features can vary based on the objectives of the website and the decisions made by the designers. Moreover, they uncovered a few illustrative examples that can assist website owners in determining whether to implement changes. Future studies should employ larger sample sizes and incorporate an analysis of user reactions to each website, which will enable us to identify the most problematic user interface features of the website (*Sasmito & Hidayattullah, 2021*).

The utilization of mobile devices is one of the methods employed to gather information. Various websites, including social networking and e-commerce platforms, have demonstrated the value of extensive data collection. However, the rise of digital technology has introduced a new challenge for online merchants, as they must now sift through vast amounts of data before applying it. The competencies in big data analytics (BDA) outline the appropriate use of social media by suppliers to increase revenue and attract new clients (*Hamid et al., 2022b*). The authors integrated multiple web analytics technologies with machine learning classifiers to gain a deeper understanding of patterns in e-commerce transaction data. Our KMP-based multivariate trimming technique was developed specifically for online index queries. Additionally, this article introduces a novel approach based on machine learning for evaluating the performance of e-commerce websites by analyzing transactional log data. To model the performance of an eLearning system concerning predictor variables, they utilized a combination of machine learning methods and multiple linear regressions (*Fatima, Ali & Ashraf, 2017*). Implementing this method can bring significant benefits to online stores. This feature allows merchants to track which products have been viewed by specific customers, enabling them to better categorize their customers and tailor their offerings accordingly. It is claimed that machine learning models' accurate predictions enhance the user experience. In today's competitive e-commerce landscape, businesses are faced with intense competition. One effective approach to enhancing value co-creation involves optimizing the user interface of an e-commerce system. To ensure continued success in supporting businesses, it is crucial to identify users' emotional criteria for success and expand user-driven standards. From a theoretical standpoint, research on user interaction is still in its early stages. This research aims to examine and validate the relationship between user involvement and technology adoption (*Riasat et al., 2023*). To achieve this, they reevaluated the standards for dependability by identifying the core features unanimously agreed upon by all parties

involved, as well as the additional features that support the concept of participation. Initially, they conducted research and analyzed the reliability of various user interactions to determine which ones would be most beneficial for online purchasing. The reliability of the constructs was found to be high, with an alpha value greater than 0.70. They identified six key features of user interaction that strongly support this point, namely aesthetic quality, persistence, novelty, participation, and concentrated attention. The process encompasses user, system, and interaction patterns. This research lays the groundwork for future studies to develop metrics for measuring the effectiveness of the user experience and conducting comprehensive analyses of the conceptual model across multiple domains (*Hussain et al., 2022a*).

In response to the COVID-19 pandemic, the authors have developed an innovative cardiovascular rehabilitation (CR) program, utilizing implementation science (ImS) and UCD methodologies. The Three-Stage Hybrid CR (THCR) program is designed to be flexible, accommodating both home and clinic settings. Through stakeholder interviews, design workshops, and journey mapping, they identified obstacles to CR adoption and created a prototype intervention. The THCR prototype leverages remote patient monitoring (RPM) technology and addresses concerns related to equipment, connectivity, patient safety, and protocol flexibility. Our theory-based, telehealth-enhanced CR program focuses on user and contextual barriers, integrating UCD and ImS approaches. They aim to assess the program's effectiveness and feasibility of implementation through a subsequent experiment (*Hamid et al., 2023*).

In light of the COVID-19 pandemic, they have devised an innovative cardiovascular rehabilitation (CR) program utilizing the principles of implementation science (ImS) and UCD. This program, known as the THCR, can be carried out both in the comfort of one's home and within clinic settings. To ensure its success, authors conducted stakeholder interviews, design workshops, and journey mapping exercises to identify any obstacles to adoption and to guide the development of a prototype intervention (*Hamid et al., 2022d*). The THCR prototype incorporates remote patient monitoring (RPM) technology and addresses concerns about equipment, connectivity, patient safety, and protocol flexibility. Our telehealth-enhanced CR program is grounded in well-established theories and places a strong emphasis on user experience and the contextual factors that may impede progress. They have integrated UCD and ImS techniques to create a comprehensive approach. The subsequent experiment will evaluate both the effectiveness and feasibility of implementation (*Hamid et al., 2022c*).

Artificial intelligence (AI) is now widely integrated into various educational domains, offering a multitude of learning aids with diverse purposes. According to the Horizon Reports of 2021 and 2022, AI will soon become a staple in every classroom and campus. This research investigates the implementation of a UCD approach in the redesigned user interface of the Blockly-Electron AI teaching program. By analyzing software usability characteristics, they derive guidelines for enhancing the intuitiveness of AI educational software in the future (*Hamid et al., 2022a*). The investigation employs attribution analysis and UCD as prominent methodologies, involving four phases of the UCD process. Seventy-three students in grades six to eight, along with five educators, participated in the

study. The USE scoring method is utilized to evaluate Blockly-Electron's potential. By establishing a linear link between ease of learning, ease of use, usefulness, and satisfaction, and using ease of use as a mediator variable, five design deliverables and an attribution model are developed (*Hamid et al., 2022a*). This study diverges significantly from previous regression studies on the USE scale, employing quantitative techniques and a UCD approach. Improved usability based on the system's functionality and learning curve will facilitate the creation of intuitive AI-assisted learning programs (*Muhammad et al., 2022*).

The internet and digital media have experienced a significant surge in popularity in recent years. This trend has had both positive and negative effects on our daily lives, leading to the rapid growth of the field of human-computer interaction (HCI). In response to increasing environmental concerns regarding the internet's impact on resource depletion, electronic waste, energy consumption, and carbon footprint, researchers and practitioners, including a subset of designers, are advocating for an eco-friendlier approach to website and application design. While this concern is relatively new, this research aims to provide guidelines for creating an intuitive smartphone interface for food delivery services. The authors examine the ordering habits and user perspectives on interfaces, with the next step being user testing on a prototype of a low-impact food delivery application (FDA) to understand how to make appealing interfaces with minimal environmental impact and ultimately influence user behavior (*Hamid et al., 2022a*). The primary objective of this study is to determine whether mobile app users place the same importance on low-impact design approaches as desktop computer users. However, regardless of screen size, the research demonstrates the benefits of incorporating low-impact features. Moreover, the general public is typically receptive to environmentally friendly innovations. Nonetheless, customers often lack the necessary expertise and understanding, which necessitates providing thorough explanations even for minor changes to an application (*Aamir et al., 2024*).

The objective of this research is to apply user experience theory to popular traditional culture apps, with the goals of (1) analyzing and understanding the key aspects of user experience within these apps, and (2) providing specialized design methodologies for their enhancement. The focus lies on evaluating how users feel when using a selection of traditional cultural apps, their interactions with the app, and the emotional responses elicited by the app (*Sajawal et al., 2022*). Additionally, a concise overview of the design procedures and approaches employed in the development of traditional cultural apps, along with their shared and unique features, advantages, and disadvantages, will be presented. It is crucial to consider user perceptions, interactions, and emotions when developing applications for ancient cultures. This entails striving for improvements in user immersion, information provision, design quality, and the establishment of a guiding model to assist users in achieving their objectives. By accomplishing these objectives, businesses can enhance customer satisfaction, foster brand loyalty, and inspire customers to remain engaged with their brand (*Bukhsh et al., 2023*).

Government-provided products and services should primarily focus on enhancing the standard of living for citizens. To ensure a satisfied population, it is crucial to establish reliable and high-quality public services. Tegal, an Indonesian city, is a prime example of

**Table 1 Comparative analysis.** A side-by-side comparison of different web-based applications regarding key usability and performance metrics.

| Ref. | Page load speed | Mobile responsiveness | Navigation | Content quality | Call-to-action (CTA) | Readability | Accessibility | Browser compatibility | Error handling | Security | Page load speed |
|---|---|---|---|---|---|---|---|---|---|---|---|
| Abdulrahman, Rawf & Gahfoor (2022) | Fast | Highly responsive | Clear and intuitive | High-quality content | Visible and compelling | Readable and structured | Accessible design | Compatible across browsers | Clear error messages | Secure (with SSL) | Fast |
| Maidom et al. (2022) | Moderate | Responsive | Somewhat complex | Good quality content | Visible | Readable | Some accessibility issues | Mostly compatible | Vague error messages | Secure | Moderate |
| Gruzdo et al. (2023) | Very fast | Highly responsive | Clear and intuitive | Exceptional content | Highly visible | Very readable | Fully accessible | Fully compatible | Clear error messages | Secure (with SSL) | Very fast |
| Sundqvist & Zurawska (2023) | Slow | Somewhat responsive | Complex | Below average content | Unclear CTA | Hard-to-read | Limited accessibility | Limited compatibility | Confusing error handling | Insecure | Slow |
| Wu et al. (2023) | Fast | Highly responsive | Clear and intuitive | High-quality content | Visible and compelling | Readable and structured | Accessible design | Compatible across browsers | Clear error messages | Secure (with SSL) | Fast |
| Sasmito & Hidayattullah (2021) | Moderate | Responsive | Somewhat complex | Good quality content | Visible | Readable | Some accessibility issues | Mostly compatible | Vague error messages | Secure | Moderate |
| Riasat et al. (2023) | Very fast | Highly responsive | Clear and intuitive | Exceptional content | Highly visible | Very readable | Fully accessible | Fully Compatible | Clear Error Messages | Secure (with SSL) | Very fast |
| Lobe, Morgan & Hoffman (2020) | Slow | Somewhat responsive | Complex | Below average content | Unclear CTA | Hard-to-read | Limited accessibility | Limited compatibility | Confusing error handling | Insecure | Slow |
| Hussain et al. (2022a) | Fast | Highly responsive | Clear and intuitive | High-quality content | Visible and compelling | Readable and structured | Accessible design | Compatible across browsers | Clear error messages | Secure (with SSL) | Fast |
| Anwyl-Irvine et al. (2021) | Moderate | Responsive | Somewhat complex | Good quality content | Visible | Readable | Some accessibility issues | Mostly compatible | Vague error messages | Secure | Moderate |

(Continued)

| Ref. | Page load speed | Mobile responsiveness | Navigation | Content quality | Call-to-action (CTA) | Readability | Accessibility | Browser compatibility | Error handling | Security | Page load speed |
|---|---|---|---|---|---|---|---|---|---|---|---|
| *Hamid et al. (2023)* | Very fast | Highly responsive | Clear and intuitive | Exceptional content | Highly visible | Very readable | Fully accessible | Fully compatible | Clear error messages | Secure (with SSL) | Very fast |
| *Hamid et al. (2022c)* | Slow | Somewhat responsive | Complex | Below average content | Unclear CTA | Hard-to-read | Limited accessibility | Limited compatibility | Confusing error handling | Insecure | Slow |
| *Riaz, Ashraf & Siddiq (2020)* | Fast | Highly responsive | Clear and intuitive | High-quality content | Visible and compelling | Readable and structured | Accessible design | Compatible across browsers | Clear error messages | Secure (with SSL) | Fast |
| *Muhammad et al. (2022)* | Moderate | Responsive | Somewhat complex | Good quality content | Visible | Readable | Some accessibility issues | Mostly compatible | Vague error messages | Secure | Moderate |
| *Ali et al. (2018)* | Very fast | Highly responsive | Clear and intuitive | Exceptional content | Highly visible | Very readable | Fully accessible | Fully compatible | Clear error messages | Secure (with SSL) | Very fast |
| *Aamir et al. (2024)* | Slow | Somewhat responsive | Complex | Below average content | Unclear CTA | Hard-to-read | Limited accessibility | Limited compatibility | Confusing error handling | Insecure | Slow |
| *Fayyaz et al. (2021)* | Fast | Highly responsive | Clear and intuitive | High-quality content | Visible and compelling | Readable and structured | Accessible design | Compatible across browsers | Clear error messages | Secure (with SSL) | Fast |
| *Bukhsh et al. (2023)* | Moderate | Responsive | Somewhat complex | Good quality content | Visible | Readable | Some accessibility issues | Mostly compatible | Vague error messages | Secure | Moderate |
| *Alsubhi et al. (2019)* | Very fast | Highly responsive | Clear and intuitive | Exceptional content | Highly visible | Very readable | Fully accessible | Fully compatible | Clear error messages | Secure (with SSL) | Very fast |

this endeavor. The Tegal city government strives to transform into a "smart city" and diligently caters to the needs of its residents in a timely, efficient, and cost-effective manner (*Ashraf, Eassa & Albeshri, 2018*). Unfortunately, the private sector and locally owned enterprises (BUMD) in Tegal have been unaware of the city government's commendable public services. To address this, a web-based planning system has been developed to assist the citizens of Tegal City in locating BUMD, public services, and private sector offerings. This article provides comprehensive information about the quickest route and a comparison of various public transportation options available in Tegal (*Mudiyono, Anindyawati & Setiawan, 2017*). Leveraging the UCD approach, software can now be developed to meet the specific requirements of individual users within a remarkable timeframe of just 105 days (or 15 weeks). The evaluation of a mapping website was conducted using the equivalence partitioning method, which yielded an overall accuracy rate of 83% for identification tests. Furthermore, on the SUS scale, designed to measure user satisfaction, a score of 89% or higher is deemed excellent (*Ashraf, 2021*). The focus of the research and its implications for both web-based and mobile applications, enhancing the overall coherence and impact of the study.

Table 1 aims to provide a side-by-side comparison of different web-based applications regarding key usability and performance metrics. This comparative analysis helps to identify strengths and weaknesses in each application and highlights best practices that can inform design and optimization efforts.

## DATA COLLECTION

To ensure comprehensive usability evaluation and optimization, a diverse group of participants was selected. Participants included individuals with expertise in various fields, such as technical communication, cognitive psychology, human factors, computer science, and usability. Their collective knowledge and experience contribute to a thorough assessment of different usability aspects. The participants were chosen based on their relevant qualifications, experience, and familiarity with web applications (*Hussain et al., 2022b*; *Ahmed et al., 2022*; *Hafeez et al., 2021*) they are described in Table 2.

The data for this research was collected through the OptiFlow framework, which operates through a web-based platform. The key data source utilized in the data collection process is The OptiFlow web service acts as a bridge between the users and the OptiFlow library. It enables users to input the necessary evaluation parameters, such as the website's title, slogan, and URL. The web service dynamically adjusts its initialization variables based on the inputs received and establishes a seamless connection with the OptiFlow library. The OptiFlow library serves as the evaluation instrument, retrieving and analyzing data from the repository. It executes the rules and guidelines defined in Rule Type 1 and the guidelines definitions in Table 3. The library interacts with the SQL tabular database to fetch the relevant information required for the evaluation process.

According to Table 4, The implementation level table defines three levels of implementation: Green, Amber, and Red. Each level is associated with a Level ID and an interpretation. The "Green" level indicates that a guideline can be fully implemented in the database within the OptiFlow framework, with conclusive results. The "Amber" level

**Table 2 Participant selection.** The participants were chosen based on their relevant qualifications, experience, and familiarity with web applications.

| P. ID | Field of expertise | Qualifications & experience | Web familiarity |
|---|---|---|---|
| 1 | Cognitive Psychology | Ph.D. in Cognitive Psychology | High |
| | | 5+ years of research in usability and user behavior | |
| | | Experience in conducting usability studies | |
| 2 | Technical Communication | M.Sc. in Technical Communication | Moderate |
| | | 3+ years of technical writing experience | |
| | | Previous involvement in usability projects | |
| 3 | Computer Science | B.Sc. in Computer Science | High |
| | | 2+ years of web development experience | |
| | | Knowledge of user interface design principles | |
| 4 | Human Factors | M.Sc. in Human Factors | High |
| | | 4+ years of experience in human factors research | |
| | | Worked on usability evaluations in the past | |
| 5 | Usability | B.A. in Human-Computer Interaction | High |
| | | 6+ years of usability testing experience | |
| | | Expertise in usability analysis and optimization | |

**Table 3 OptiFlow library.** The rules and guidelines defined in the rule Type 1 and the guidelines.

| Component | Description |
|---|---|
| Function | The OptiFlow library serves as the evaluation instrument, responsible for retrieving and analyzing data from the repository. |
| Execution | The library executes the defined rules and guidelines from the Rule Type 1 table and the Guidelines Definitions table. |
| Interaction | It interacts with the SQL tabular database to fetch relevant information required for the evaluation process. |
| Role | The OptiFlow library plays a central role in the framework, as it is responsible for conducting the usability evaluation based on the provided rules and guidelines. |
| Data retrieval | The library retrieves the necessary data from the SQL tabular database to obtain rule type properties and guideline definitions. |
| Rule execution | It executes the rules defined in the Rule Type 1 table to evaluate website usability based on HTML and CSS elements. |
| Guideline analysis | The library analyzes the guidelines from the Guidelines Definitions table to assess the compliance of websites with usability principles. |
| Data analysis | With the collected data, the library performs data analysis to identify usability violations and prioritize critical usability concerns. |
| Result generation | After the evaluation process, the library generates evaluation results, which are then passed on to the web service for further processing. |

suggests that the guideline is harder to implement, but additional AI algorithms can potentially convert it into a "green" guideline. The "Red" level signifies that the guideline is abstract and may require advanced algorithms or user intervention to be implemented within the framework.

The data analysis process involves a systematic evaluation of website usability based on the OptiFlow framework's evaluation results. The data collected from the web service is

**Table 4 Implementation level table.** Implementation level table defines three levels of implementation: Green, Amber, and Red. Each level is associated with a Level ID and an interpretation.

| ID | Levels of implementation | Interpretation |
|----|--------------------------|----------------|
| 1 | Green | • The OptiFlow framework allows seamless integration of guidelines into the database, ensuring their full implementation.<br>• Through its advanced capabilities, the framework can automatically assess whether a given guideline is relevant to the specific website under evaluation.<br>• The framework consistently provides definitive results when referencing such guidelines, as they are usually measurable and come with well-defined parameters. |
| 2 | Amber | • The seamless integration of the guideline into the OptiFlow framework presents some challenges.<br>• To address this, specific patterns have been integrated into the database, enabling automatic identification of the guideline's relevance to the website under evaluation.<br>• To transform this guideline into a more effective and user-friendly "green" guideline, OptiFlow can leverage additional Artificial Intelligence algorithms.<br>• The output generated by the OptiFlow framework provides valuable data to aid human evaluators in determining the applicability of the guideline to the website they are assessing. |
| 3 | Red | • This guideline is inherently abstract and demands either user intervention or the implementation of highly advanced algorithms from the field of Artificial Intelligence, or the integration of additional technology within the framework to make it viable for implementation.<br>• By leveraging sophisticated algorithms or cutting-edge technology, it can be transformed into an "amber" or "green" guideline.<br>• Currently, the framework includes this guideline in a way that requires human evaluators to manually verify its applicability to the website being evaluated. |

**Table 5 Website usability data analysis for guidance 81.** The data collected from the web service is organized into a data table, including fields such as guideline definitions.

| Guideline identifier | Rule type properties | Results | Fields |
|----------------------|----------------------|---------|--------|
| Guideline 81 | Rule Type 1 properties | Tags: 3<br>Fail: 1<br>Success%: 66.7 | Success: 2<br>Null: 0<br>Passed: FALSE |

organized into a data table, including fields such as guideline definitions, rule type properties, and results fields. The analysis is performed as follows in Table 5.

The data collected from the web service is visualized using data views, which are created for each implementation level—"Green," "Amber," and "Red." These data views present the evaluation results in a clear and organized manner, displaying usability violations and their priority ratings.

Table 6 presents the data visualization of usability violations categorized into different implementation levels: "Green," "Amber," and "Red." Each violation is associated with a corresponding Priority Rating, indicating its relative importance. The data views offer a clear and organized representation of usability concerns, enabling users to identify and prioritize critical violations for effective optimization.

**Table 6 Data visualization—usability violations.** Presents the data visualization of usability violations categorized into different implementation levels: "Green," "Amber," and "Red." Each violation is associated with a corresponding Priority Rating, indicating its relative importance.

| Implementation level | Usability violation | Priority rating |
|---|---|---|
| Green | Violation 1 | 4 |
| Green | Violation 2 | 3 |
| Green | Violation 3 | 2 |
| Green | Violation 4 | 2 |
| Amber | Violation 5 | 5 |
| Amber | Violation 6 | 4 |
| Amber | Violation 7 | 3 |
| Red | Violation 8 | 5 |
| Red | Violation 9 | 5 |

## METHODOLOGY

This research introduces a novel approach to streamline usability evaluation and optimization known as OptiFlow. Among various techniques, heuristic evaluation emerges as the most effective method for identifying prevalent usability issues by assessing the interface against established usability principles. The goal of an online application's instructional design is to enable users to effortlessly and rapidly complete tasks. Leveraging the OptiFlow framework, objective assessments of a website's usability can be conducted, guided by well-defined criteria. This technical advancement promises to enhance the overall user experience and optimize digital interfaces for maximum efficiency and user satisfaction.

The many components of the OptiFlow architecture are depicted in Fig. 1.

**User:** An individual responsible for assessing the usability of a website.

**Web site:** The OptiFlow system operates through a web-based platform, enabling users to engage with it through various interfaces. To access the desired viewing page, the user needs to input the website's title, slogan, and URL.

**Web service:** The web service establishes a seamless connection with the library by employing the execute method, where essential analysis parameters are passed. Upon completion, the online service promptly delivers the review results back to the website.

**Library:** The evaluation instrument is readily available within the library, efficiently retrieving and analyzing data from the repository.

**Database:** There are four tables in the SQL tabular database.

**Usability category table:** Table 7 presents a comprehensive categorization of the various forms of usability.

**Implementation level table:** The "Implementation Level" table in the Data Collection provides a detailed overview of the different types of implementations.

**Guidelines definitions table:** The "Guidelines Definitions" table offers readers a clear and straightforward explanation of all the framework guidelines. Each priority

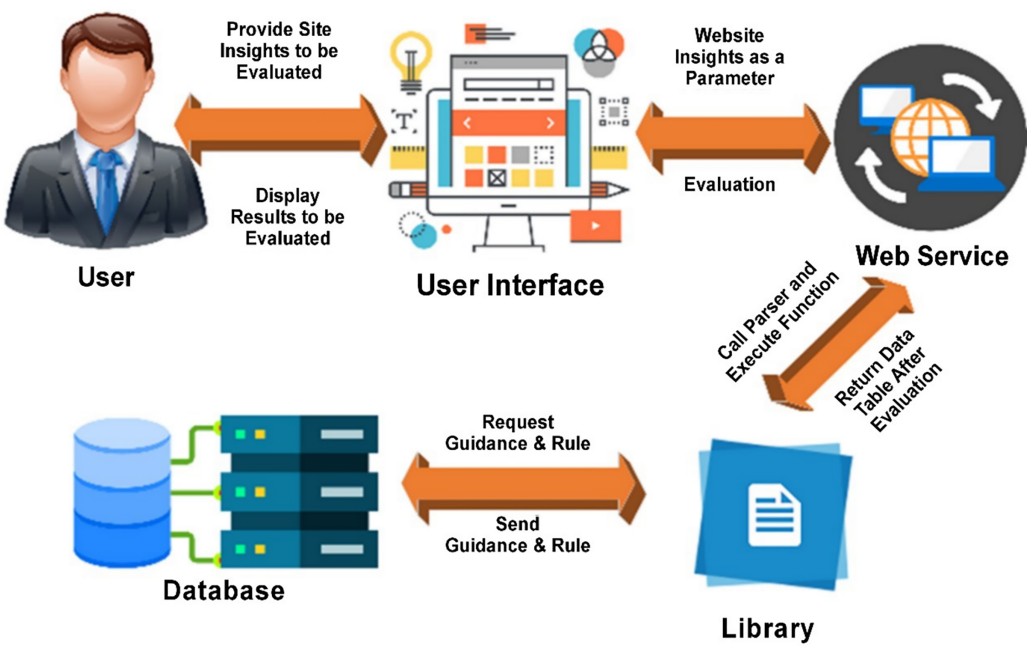

**Figure 1  Architecture of OptiFlow.**     

classification (Data Collection) and corresponding table, outlining the standard's usability and execution level, are hyperlinked below for easy access.

**Rule Type 1 table:** This table is a summary of the "green" and "amber" implementation level requirements from the guidelines' definitions table (Data Collection). Each record has fields for the HTML element and any other information the library needs to locate the pattern corresponding to a given rule, such as the attribute and size of the element. The rule type 1 table can be used to locate and compare two HTML tags or tags contained within other tags. Significant integers indicating whether the rule has been broken or not are included in the "ruleSuccess" column's elements. In terms of website usability, it's vital to differentiate between rules that should be accessible at all times and regulations that should only be accessed once. To help the library locate rules that apply to HTML elements and CSS filters, the current OptiFlow framework only has a single rule type, the Rule Type 1 table. Additional rule categories, such as those that guide the library in determining the most effective use of photos and other digital resources, will be included in subsequent printings. If further structural tools, like a JavaScript parser, are developed, the system will need new rule types to accommodate them.

Over the years, several usability guidelines have been formulated, but they have not reached a unanimous consensus as there is no dominant set of recommendations. For this project, experts and researchers from various fields such as cognitive psychology, technical communication, computer science, human factors, and usability conducted extensive usability tests. They collected valuable data from these tests and leveraged it to create 240 principles. All the proposed criteria were then organized following the HHS Research-Based Web Design & Usability guidelines. Table 7 showcases the 15 categories into which each recommendation was classified.

**Table 7 Guidance against each category.** Showcases the 15 categories into which each recommendation was classified.

| Usability category | Number of guidelines |
| --- | --- |
| Optimizing the user experience | 29 |
| Hardware and software | 4 |
| The homepage | 12 |
| Page layout | 9 |
| Navigation | 27 |
| Scrolling and paging | 3 |
| Headlines, titles and labels | 18 |
| Links | 21 |
| Text appearance | 18 |
| Lists | 13 |
| Screen based controls (widgets) | 27 |
| Graphics, images and multimedia | 17 |
| Writing web content | 18 |
| Content organization | 8 |
| Search | 16 |
| **Total guidelines** | **240** |

The OptiFlow framework indicates whether a rule can be transformed into a program-compatible format at the Implementation Level. Table 8 below provides a breakdown of the percentage of each regulation that can be implemented independently.

Unclear usability standards can pose challenges during their implementation. To address this issue, the OptiFlow framework has undergone updates, including the incorporation of rule type 1 tables and recommendation definitions (Data Collection). These enhancements aim to alleviate the difficulties associated with implementing the standards. As an illustration of the process, the study utilize OptiFlow Framework Guideline #81 to demonstrate how to incorporate a guideline into either of these tables.

To enhance memorability and ease of typing, it is recommended to keep URLs under 50 characters. By adhering to this guideline, both Search Engine Optimization (SEO) and usability aspects of the website will experience positive impacts.

The primary key for the plan in the rules database is 81 as shown in Table 9. The "Guideline" section contains the rules, while the "Reason" section provides the rationale behind each rule. To assess this recommendation as per Rule Type 1, ensure that Rule Type 1 is provided in the "ruleType" parameter. The "ruleCat" column has a value of "1," indicating that this rule is categorized as "green," signifying its compliance or positive status. Furthermore, the "ruleSeverity" column holds a value of "5," indicating a "Priority Rating" of 5 for this rule, implying its high importance. To locate the number 5 in the field immediately below the "ruleGroup" column, refer to the "navigation" usage category, as this criterion falls under it.

Table 8 **Interpretation of guidance in terms of their implementation category.** Provides a breakdown of the percentage of each regulation that can be implemented independently.

| Levels | Interpretation |
|---|---|
| Green | • The framework of OptiFlow allows for the comprehensive implementation of guidelines within a database. |
| | • The framework possesses the capability to autonomously ascertain whether the presently loaded page satisfies this criterion. |
| | • Due to the ability to precisely quantify these principles within clearly defined contexts, the system adhering to this guideline consistently produces outcomes that may be anticipated. |
| Amber | • The challenging component lies in the successful implementation of the suggested adjustment proposed by the OptiFlow design. |
| | • The database has been trained to recognise specific patterns that indicate whether or not the website being scanned comes into the aforementioned category. |
| | • The environmental sustainability of this guidance could be enhanced by including more artificial intelligence methodologies within the OptiFlow framework. |
| | • When the guideline is included in the framework's results, the evaluator can use the provided context to assess its relevance to the website being evaluated. |
| Red | • The aforementioned criterion is commonly conceptualized as being dependent on the utilization of human agents, very advanced AI algorithms, or other technical methods |
| | • For its actualization. The criteria have the potential to be modified to fall within either the "amber" or "green" categories through the utilization of sophisticated mathematics or advancements in technology. |
| | • The existing iteration of the framework incorporates a certain criterion that can be subject to manual scrutiny in order to determine its applicability to the website under consideration. |

Table 9 **Representation of guidance 81 in guidance table.** To locate the number 5 in the field immediately below the "ruleGroup" column, refer to the "navigation" usage category, as this criterion falls under it.

| pk | Guideline | Reason | Rule type | Rule cat | Rule severity | Rule group |
|---|---|---|---|---|---|---|
| 81 | URLs should not be complex | URLs should ideally be less than 50 characters. Such URLs are beneficial for both usability and SEO | 1 | 1 | 5 | 5 |

In this section, the study outline the post-visit process for users of OptiFlow, covering the necessary steps after entering the assessment parameters. This aims to give a comprehensive overview of the components depicted in Fig. 1 and their interactions to facilitate the display of usability evaluation results.

Subsequently, the user informs the OptiFlow framework about their desired webpage. This can be accomplished by providing the website's name, URL, and optionally, the catchphrase in the provided blanks. Upon completing the form in Step 1, the user can submit the provided information as evaluation parameters to the online service by simply clicking the "Evaluate Website" button.

The web service initiates communication with the library, dynamically adjusting its initialization variables based on the inputs received from the website. Both the library and the web service play integral roles in Fig. 1, with distinct functions that are visually highlighted. It is important to note that the term "communication" between the library and web service refers to a shared understanding of their interaction. In this context, the web host is responsible for determining the company's name, slogan, and web address.

**Table 10 Data table structure stored.** Data that has been returned by a web service is saved using Table 5 as a guide.

| Guideline definition | Rule type properties | Results fields |
| --- | --- | --- |

Subsequently, the web service utilizes the library to access and integrate the website, resulting in parsed HTML and CSS documents. These files are then stored in static variables by the web service.

To retrieve data from the tables for the various types of rules and suggestions, the web service interacts with the library, which then interacts with the SQL database. Data that has been returned by a web service is saved using Table 5 as a guide presented in Table 10.

Sometimes, it makes things easier to have a blank table because at least you have a format where you can put your information into. Table 10 declares the categories or variables before data is collected leading to systematic arrangement of the collected data.

**1 data row**

The components of the data table shown in Table 5 are the following:

**Guideline definition:** This section presents a comprehensive restatement of the information found in the Guidelines Definitions table, as outlined in the Data Collection section.

**Rule type properties:** Within this column, you will find a replication of the table that outlines the attributes of Rule Type 1.

**Results fields:** "Results fields" refer to additional fields integrated into the data table of the web service without any initial values. Once processing is finalized, the web service will populate these fields with the discoveries made and transmit them back to the host server.

These are:

**Tags:** This field will indicate the frequency of occurrence of HTML tags or CSS selectors found in the column under the "Rule Type Properties" in each data row.

**Success:** It represents the count of tags or selectors that match the properties specified in the referenced guideline.

**Fail:** The number of tags or selectors whose attributes match the properties of the guideline being referenced, but their value or size properties do not match.

**Null:** The count of tags or selectors that do not match the property of the guideline in terms of attributes, sizes, or properties.

**Success%:** This value is calculated using the formula: $Success/(Tags - Null) \times 100$.

**Passed:** Eventually, this field will hold a True/False value, indicating whether the guideline has been adhered to or violated.

After completing Step 4, the web service's static variables will hold copies of the HTML and CSS files, along with the information from Table 5. The initial data row returned by the web service is then passed on to the library execute method. Table 5 demonstrates that the execution procedure only requires a single row of input data. Upon receiving a data

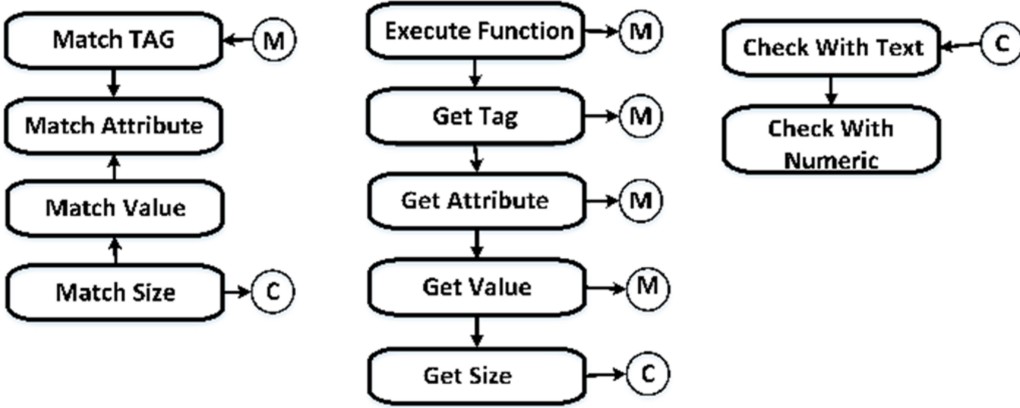

**Figure 2** Logic base tree for rule Type 1.               

**Table 11 Comparison by execute function for guidance 81.** Rule comparisons may manifest in the following manner.

| Input A (Evaluation result) | Input B (ruleSuccess) | Output (A XNOR B) |
| --- | --- | --- |
| True | True | True: Increment success counter |
| True | True | True: Increment success counter |
| False | True | False: Increment fail counter |

row from the web service, the library execute function determines which rule to apply to assess whether the website complies with the specified usefulness criteria. For the sake of explanation, it will assume the presence of Rule 81. When the execute method of the web service is invoked with a record containing the "a" identifier, the logic tree depicted in Fig. 2 will be executed to ensure that Rule #81 is not violated. Figure 2 showcases the versatility of the execute function, as it can take various forms depending on the data in the fields and the level of pattern matching required.

The run function's return value is contingent upon the test's conclusion, which may be True, False, or null. Directive 81 provides definitions for the subsequent terms:

**True:** The "href" property of the "a" element has a maximum length of 70 characters.

**False:** The statement lacks veracity. The recommended minimum length for the "href" attribute of the "a" element is 70 characters.

**Null:** The "a" element does not provide support for the "href" attribute.

Considering Principle #81, delineates the specific circumstances in which the execute function yields two True, one False, and one null evaluation outcomes. Consequently, rule comparisons may manifest in the following manner presented in Table 11.

The finalized data table is transmitted to OptiFlow using a web API. To enhance the visual representation of the data, OptiFlow generates three distinct visualizations, each corresponding to a specific execution tier. The system effectively informs the user about the breaches associated with each application, categorizing them in descending order of severity.

# RESULTS AND DISCUSSION

In the fast-paced digital age, where user experience is paramount, the OptiFlow framework emerges as a cutting-edge solution to enhance the usability and optimization of online applications. By seamlessly integrating heuristic evaluation and a web-based platform, OptiFlow redefines the landscape of usability assessment, aiming to elevate user experiences and drive digital interface efficiency to new heights.

The underlying architecture of OptiFlow, vividly depicted in Fig. 1, represents a thoughtfully orchestrated framework that seamlessly facilitates the process of usability evaluation and optimization. This architecture orchestrates the harmonious interaction among pivotal components—the "User," the "Site," the "Web Service," and the "Library"– culminating in a cohesive and comprehensive usability assessment. The architecture's design is inherently geared towards enabling smooth data exchange, resulting in precise and accurate evaluations anchored in the parameters provided.

At the heart of the OptiFlow framework lies a comprehensive set of 240 usability guidelines, meticulously categorized into 15 distinct usability categories. These guidelines are not arbitrary; they are derived from extensive usability tests and insights contributed by experts spanning cognitive psychology, technical communication, computer science, human factors, and usability domains. The framework's methodology echoes alignment with the well-established HHS Research-Based Web Design & Usability guidelines, thereby substantiating the credibility and robustness of the usability assessment process.

The strength of the OptiFlow framework lies in its methodical approach to categorizing guidelines into three distinct implementation levels: "Green," "Amber," and "Red" as shown in Table 12. This hierarchical classification empowers the framework to prioritize tasks. "Green" guidelines represent the realm of the readily implementable, characterized by well-defined parameters and quantifiable outcomes. Contrarily, "Amber" guidelines, often entailing intricate nuances, necessitate the incorporation of AI algorithms for successful execution. The "Red" category encapsulates abstract and intricate concepts, often requiring advanced technologies or specialized user interventions for effective integration.

This table succinctly outlines the characteristics of each implementation level within OptiFlow. It delineates the strengths and complexities associated with "Green," "Amber," and "Red" guidelines, enabling stakeholders to discern the appropriate approach for each usability challenge.

The usability evaluation process within OptiFlow is a well-defined and robust endeavor. The cornerstone of this process is the indispensable "execute" function, responsible for meticulously assessing a website's adherence to specific usability guidelines. This function processes the website's raw data, strategically applies pre-established logic trees, and subsequently yields outcomes of "True," "False," or "Null." These outcomes, bearing the imprint of the website's usability against the stipulated guidelines, are then juxtaposed with guideline parameters. This process culminates in a systematic increment of counters for "Success," "Fail," and "Null" within the framework's data table.

**Table 12 Categorization and implementation levels in OptiFlow framework.** Outlines the characteristics of each implementation level within OptiFlow.

| Implementation level | Description | Characteristics |
| --- | --- | --- |
| Green | Easily implementable | Well-defined parameters and measurable outcomes. |
| | | Guidelines that can be integrated without complex AI algorithms. |
| | | Quantifiable and demonstrable results. |
| Amber | Moderately complex | Guidelines with nuances requiring AI algorithm integration. |
| | | May involve patterns and trends recognition. |
| | | Improved integration with additional AI technology. |
| Red | Highly complex | Abstract and complex guidelines. |
| | | May require advanced AI algorithms or specialized user interventions. |
| | | Often entails non-standard usability challenges. |

While the OptiFlow framework already stands as an exemplar of innovative usability evaluation, it harbors promising avenues for future advancements. These encompass the potential incorporation of additional rule types to address a broader spectrum of website elements and resources. Furthermore, the infusion of advanced technologies, such as JavaScript parsers, holds the potential to further expand the framework's capabilities. This would empower OptiFlow to tackle intricate and nuanced usability challenges with precision and efficacy. In summation, the OptiFlow framework maps an evolution in the realm of usability evaluation and digital interface optimization. The unity of its architecture's coherence, the employment of meticulously formulated usability guidelines, the systematic categorization of guidelines, the structured integration into an SQL database, the methodical usability evaluation process, and the intelligent results presentation collectively converge to usher in a new era of enriched user experiences and finely-tuned digital interfaces.

The OptiFlow framework represents more than just a methodology; it's a testament to the ever-evolving landscape of usability evaluation and optimization. Its architecture, methodology, and innovative approach collectively usher in a holistic assessment of digital interfaces. By harmonizing expert-sourced guidelines, methodical categorization, and meticulous evaluation mechanisms, OptiFlow empowers stakeholders to proactively sculpt user experiences and fine-tune the operational efficiency of digital interfaces. As digital horizons continue to evolve, the OptiFlow framework stands as a beacon of ingenuity, steering the course of innovative usability assessment and optimization toward uncharted territories.

## CONCLUSIONS

In conclusion, the OptiFlow framework emerges as a pioneering solution to streamline usability evaluation and optimization, marking a significant advancement in the realm of digital interface enhancement. Through a meticulously orchestrated architecture, expert-derived guidelines, methodical categorization, structured integration into an SQL database, and a systematic evaluation process, OptiFlow empowers stakeholders to proactively

address usability challenges and elevate user experiences. The architecture's efficacy lies in its seamless orchestration of essential components, enabling a cohesive usability assessment process. This methodology's foundation is laid upon a comprehensive set of 240 usability guidelines, backed by insights from diverse domains. The guidelines are thoughtfully categorized into "Green," "Amber," and "Red" levels, facilitating the prioritization of usability challenges and providing a roadmap for effective interventions. The structured integration of guidelines into the SQL database exemplifies the framework's commitment to clarity and ease of implementation. By utilizing the execute function, the framework meticulously evaluates websites against guidelines, yielding outcomes that are then juxtaposed against predefined parameters. The results presentation, categorized and prioritized, equips stakeholders with actionable insights to address high-priority issues swiftly. OptiFlow's innovative approach does not end with its current state; it holds the potential for further advancement. The inclusion of additional rule types and advanced technologies like JavaScript parsers promises to extend the framework's capabilities, addressing a broader array of usability challenges. As the digital landscape continues to evolve, the OptiFlow framework stands as a beacon of innovation, driving the narrative of usability assessment and optimization forward. Its comprehensive approach, systematic evaluation, and structured results presentation herald a new era of enriched user experiences and finely tuned digital interfaces. Ultimately, OptiFlow represents a pivotal step toward harmonizing the intricate balance between human-centric design and technological sophistication in the ever-evolving digital realm.

### Funding
Qassim University, represented by the Deanship of Graduate Studies and Scientific Research, financially supported this research under the number (QU-APC-2024-9/1) during the academic year 1445 AH/2023 AD. The funders had no role in study design, data collection and analysis, decision to publish, or preparation of the manuscript.

### Grant Disclosures
The following grant information was disclosed by the authors:
Qassim University.
Deanship of Graduate Studies and Scientific Research: QU-APC-2024-9/1.

### Competing Interests
The authors declare that they have no competing interests.

### Author Contributions
- M. Waseem Iqbal conceived and designed the experiments, performed the experiments, analyzed the data, performed the computation work, authored or reviewed drafts of the article, and approved the final draft.
- Khlood Shinan conceived and designed the experiments, analyzed the data, prepared figures and/or tables, and approved the final draft.

- Shahid Rafique Shahid Rafique conceived and designed the experiments, performed the experiments, analyzed the data, prepared figures and/or tables, authored or reviewed drafts of the article, and approved the final draft.
- Abdullah Alourani conceived and designed the experiments, performed the experiments, analyzed the data, authored or reviewed drafts of the article, and approved the final draft.
- M. Usman Ashraf conceived and designed the experiments, analyzed the data, prepared figures and/or tables, and approved the final draft.
- Nor Zairah Ab Rahim analyzed the data, performed the computation work, prepared figures and/or tables, authored or reviewed drafts of the article, and approved the final draft.

## Data Availability

The raw data is available in the Supplemental File.

## Supplemental Information

Supplemental information for this article can be found online at http://dx.doi.org/10.7717/peerj-cs.2561#supplemental-information.

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
