# Peer review of "Usability and optimization of online apps in user’s context"

_PeerJ Computer Science, doi:10.7717/peerj-cs.2561_

## Round 0.1 · original submission · Major Revisions

Dear authors,

Thank you for the submission. The reviewers’ comments are now available. We advise you to revise the paper in light of the reviewers’ comments and concerns before resubmitting it. The followings should also be addressed:

1. Pay special attention to the usage of abbreviations. Spell out the full term at its first mention, indicate its abbreviation in parenthesis and use the abbreviation from then on.
2. Some paragraphs are too long to read. They should be divided into two or more for readability and comprehensibility.
3. All references in a reference list need to be cited at least once in the text. Each work cited in the text must appear in the reference list. "References" section lists 36 references although there are 49 references in the manuscript.

Best wishes,

·

Basic reporting

sufficient background and demonstration done

Experimental design

Article defines the relevant and meaningful result analysis

Validity of the findings

The data validates the findings

Additional comments

The conclusion is appropriate and supported by the result .

Cite this review as

Reviewer 2 ·

Basic reporting

The paper details a framework and software to optimize the usability of online applications. The proposal is novel and interesting, and the description of the tool is duly detailed. However, the validation of the proposal itself has not been found and this is the main problem of the article, as will be detailed later.
The structure of the paper can be improved for a better understanding of the research.
The English used throughout the article is professional and clear. However, some statements may not be supported by data.
The intro and background are detailed; however, they are presented as a Literature Review, which may not be fulfilled, as will be indicated below.

Experimental design

It is not clear what the experimental design was, what design was followed in the research, or how the questions posed in the article have been answered.

Validity of the findings

The findings with which it is concluded are not properly supported by data, since how the proposal has been submitted to validation is not found.

Additional comments

Table 1 is not mentioned in the text, and the lack of a heading makes it difficult to identify what is being referred to or compared.
Sections 3 and 4 have the same name
Section 2 on Literature Review does not seem to be one. A Literature Review must follow a previously established protocol and meet quality criteria, such as those of Prisma 2020. Revise either the way the information is presented or the section's name.
There is no clear understanding of user interaction with the system. A user journey map or similar tool would be appreciated.
The results and discussion section needs to clearly present the results. How was the proposal evaluated? what results were obtained? how does it answer the research questions posed at the beginning?
The conclusions section does not conclude based on data. I think that understanding how each research question is answered and how the proposal was finally evaluated would be appropriate.

Cite this review as

·

Basic reporting

No comment

Experimental design

Research questions are Not well defined.
Method are not explained well.

Validity of the findings

Because of the issues of the research questions and methods, this section is hard to be assessed.

Additional comments

I have attached my comments.

Cite this review as

---

## Round 0.2 · accepted · Accept

Dear Authors,

I am grateful for your revised paper. One of the reviewers from the previous round had already accepted your paper. One of the other two reviewers did not respond to the invitation to re-review. Finally, Reviewer 3 believes that you have made the necessary additions and modifications. I concur with this assessment and believe that your paper has been sufficiently improved and is now suitable for publication.

Best wishes,

·

Basic reporting

Yes.

Experimental design

Yes.

Validity of the findings

Yes.

Additional comments

NA

Cite this review as